# RNA Interference: Promising Approach to Combat Plant Viruses

**DOI:** 10.3390/ijms23105312

**Published:** 2022-05-10

**Authors:** Sehrish Akbar, Yao Wei, Mu-Qing Zhang

**Affiliations:** 1Guangxi Key Laboratory for Sugarcane Biology & State Key Laboratory for Conservation and Utilization of Agro Bioresources, Guangxi University, Nanning 530005, China; sehrishakbar2050@yahoo.com (S.A.); yaoweimail@163.com (Y.W.); 2IRREC-IFAS, University of Florida, Fort Pierce, FL 34945, USA

**Keywords:** gene silencing, RNA interference (RNAi), host-induced gene silencing (HIGS), virus-induced gene silencing (VIGS), spray-induced gene silencing (SIGS)

## Abstract

Plant viruses are devastating plant pathogens that severely affect crop yield and quality. Plants have developed multiple lines of defense systems to combat viral infection. Gene silencing/RNA interference is the key defense system in plants that inhibits the virulence and multiplication of pathogens. The general mechanism of RNAi involves (i) the transcription and cleavage of dsRNA into small RNA molecules, such as microRNA (miRNA), or small interfering RNA (siRNA), (ii) the loading of siRNA/miRNA into an RNA Induced Silencing Complex (RISC), (iii) complementary base pairing between siRNA/miRNA with a targeted gene, and (iv) the cleavage or repression of a target gene with an Argonaute (AGO) protein. This natural RNAi pathway could introduce transgenes targeting various viral genes to induce gene silencing. Different RNAi pathways are reported for the artificial silencing of viral genes. These include Host-Induced Gene Silencing (HIGS), Virus-Induced Gene Silencing (VIGS), and Spray-Induced Gene Silencing (SIGS). There are significant limitations in HIGS and VIGS technology, such as lengthy and time-consuming processes, off-target effects, and public concerns regarding genetically modified (GM) transgenic plants. Here, we provide in-depth knowledge regarding SIGS, which efficiently provides RNAi resistance development against targeted genes without the need for GM transgenic plants. We give an overview of the defense system of plants against viral infection, including a detailed mechanism of RNAi, small RNA molecules and their types, and various kinds of RNAi pathways. This review will describe how RNA interference provides the antiviral defense, recent improvements, and their limitations.

## 1. Introduction

The world population is increasing rapidly and is expected to be double within the next 50 years, creating a massive demand for worldwide food production [1]. One way to increase food production is to develop new crops with an enhanced resistance to various biotic stress factors, such as viruses, bacteria, and fungi, which are currently a significant threat to agriculture.

Viruses are infectious pathogens, and plant virus epidemics can have devastating consequences to crop yield and quality. Every year viruses cause damage to the quality and quantity of crops. It is difficult to measure the actual losses caused by viruses worldwide, but according to an estimate, about $60 billion in losses were estimated to be caused by viral diseases [2]. For example, *Potato leaf roll Polerovirus* caused annual losses of about $100 million in the USA [2,3]. Moreover, *Barley yellow dwarf virus* (BYDV) damaged a variety of cereals such as rice, barley, maize, and oat and caused an economic loss of about $12 million per year [2,4]. The cassava crop is one of the staple crops for more than 500 million people globally. However, *Cassava mosaic begomovirus* caused annual losses of 25 million tons in Sri Lanka, Africa, and India, which often cause famine in these countries [5,6]. Moreover, the *sugarcane mosaic virus* affects the sugarcane crops every year and causes up to 20% losses when the disease incidence reaches 50% [7].

## 2. Development of Resistance against Viral Infection Inside the Host Cell

Plant viruses are a significant threat to crops and cause approximately 10–15% economic loss [8]. To encounter the virus attack, the host comprises various barriers, including (i) physical means such as cuticles, wax deposition, and the thickening of the cell wall, which assist in the blockage of virus transmission through insect vectors, (ii) passive resistance in which the host deprives the virus of vital components required for its life cycle, (iii) pattern-triggered immunity, (iv) effector-triggered immunity, (v) an RNA-silencing mechanism [9].

### 2.1. Pathogen-Triggered Immunity (PTI) against Virus Infection

Pathogen-triggered immunity is initiated by two different types of pattern recognition receptors (PRR), which are a receptor-like protein (RLP) and a receptor-like kinase (RLK) [10]. PTI triggers an influx of calcium ions (Ca+), mitogen-activated protein induction, the production of reactive oxygen species (ROS), salicylic acid production and signaling, nitric oxide synthesis, and cell wall strengthening [11,12]. PRR detects a microbe-associated molecular pattern (MAMP) [10], a pathogen-associated molecular pattern (PAMP) [13,14], and a damage-associated molecular pattern (DAMP) [13].

Upon virus infection, the PTI of the host cell is initiated, which confers immunity against the virus infection. An intricate signaling network of cytoplasmic kinase receptors and co-receptors, such as Somatic Embryogenesis Receptor Kinases (SERKs) and LRR-RLK brassinosteroids Insensitive1 (BRI1)-Associated Kinase 1 (BAK1/SERK3), are involved [15]. Similar to DAMP, viral PAMPs, known as VAMPs, are activated to resist viral infection [16]. However, some viruses surpass this defense system by encoding suppressors for PTI. These reported viruses are the *Cauliflower mosaic virus* (CaMV), *Rice stripe virus* (RSV), and *Plum pox virus* (PPV) [17,18,19].

### 2.2. Effector-Triggered Immunity (ETI) against Virus Infection

ETI has been considered the second line of the defense system. The response is initiated when effector proteins (released by plant pathogens that suppress the PTI response) are recognized by ETI [13]. Programmed cell death (PCD) is commenced at the site of the infection to inhibit the virus’ spread. The intracellular immune response is related to nucleotide-binding/Leucine-rich repeat (NLR) proteins [20,21]. NLR proteins are divided into two categories: a toll-like interleukin-1 receptor (TIR) and an N-terminal coiled-coil, referred to as TNLs or CNLs [22]. Previous studies suggest that CNLs formed a resistosome complex (calcium–permeable ion channel), which directly activates ETI [21,23]. However, TNLs are activated with NADase activity, which converts into a holoenzyme. The complexes trigger Enhanced Diseased Susceptibility (EDS1) and are facilitated by helper NLRs, named RNLs [24,25,26,27]. ETI provides resistance against virus infection through hypersensitive response (HR), which is enhanced by phytohormone production [28]. ETI provides local immunity at the site of infection, but it also provides resistance against systemic infection, classified as systemic acquired resistance (SAR) [9].

### 2.3. Gene Silencing

Gene silencing is the natural defense system in eukaryotes which provides resistance against a virus infection and regulates endogenous gene expression [29,30,31]. The initial concept of gene silencing was given by Sanford and Johnston when they reduced the pathogenicity inside plants by expressing the core genetic constituents of pathogens [29]. There are two types of gene silencing inside plant cells that ultimately downregulate the transcript level. The first occurs at a transcriptional level known as transcriptional gene silencing (TGS). In contrast, the second type of silencing is post-transcriptional gene silencing (PTGS), which degrades the accumulated mRNA in cells [32].

#### 2.3.1. Transcriptional Gene Silencing (TGS)

TGS is the form of epigenetics gene silencing that causes the methylation of promoter DNA sequences. It produces a heterochromatin area around the gene which is ultimately responsible for gene transcription repression [33,34].

#### 2.3.2. Post-transcriptional Gene Silencing (PTGS)/RNA Interference (RNAi)

Post-transcriptional gene silencing is the RNA interference (RNAi) phenomenon. With the advent of technology, the RNA silencing mechanism has been identified in plants [35]. Subsequently, RNA silencing/gene silencing has been utilized to generate successful resistance against various plant viruses. RNAi is triggered by two types of small RNA that are microRNAs (miRNAs) and small interfering RNAs (siRNAs). These small RNAs are produced by the precursor of a stem-loop structure or through long double-stranded RNA [36,37]. The siRNA and miRNA silencing phenomena follow the same biochemical pathway inside the cell.

## 3. General Mechanism of RNA-Induced Gene Silencing

Gene silencing is triggered by small RNA (sRNAs) molecules. sRNAs are small interfering RNA (siRNA) and microRNA (miRNA), depending on the origin and processing. The silencing process is initiated when long dsRNAs are cleaved into small fragments (21–25 nts) of sRNAs with the aid of the Dicer (DCL) enzyme inside the cytoplasm [38,39,40]. Next, these sRNAs duplexes unwind themselves and are loaded into the RISC. The thermodynamic stability determines the polarity of the strands as the 5’ end of the less stable strand is selected as the guide strand, and the second strand is the passenger strand, which is later discarded. Guide strands integrate into the RISC and activate it. An essential member of the RISC is the Argonaute (AGO) protein. After loading the guide strand, the AGO protein, by its endonucleolytic action, mediates the repression or degradation of the targeted mRNA [41,42]. Cleavage of targeted mRNA is initiated from 10–12 nts away from the region’s center, recognized by the guide sRNA [39]. The mechanism is accomplished by amplifying sRNA molecules through RNA-dependent RNA polymerase (RDRs) enzymes. RDRs produced double-stranded RNA, which are further cleaved and processed by DCLs and continued in the next round of RNA silencing [43] (Figure 1).

### 3.1. Enzymes Involved in RNA Induced Gene Silencing

#### 3.1.1. Dicer

Dicer is a multiple-domain protein about ~220-kDa in size, which belongs to the RNase III class and generates sRNA from long dsRNAs (using ATP). Dicer comprises of an N-terminal RNA helicase C/ATPase domain, two RNase III domains (RNase III a, RNase III b), a Piwi-Argonaute-Zwille (PAZ) domain, a domain of unknown function 283 (DUF283), and dsRNA-binding domains (dsRBDs) [44,45]. Helicase unwinds the duplex with the aid of energy produced by hydrolyzing the nucleotide triphosphate. The PAZ domain is responsible for recognizing the end of dsRNA, while RNase III domains catalyze the cleavage of dsRNA [46]. After cleavage, dicer generates two nt overhang at the 3’ region in each strand [46,47]. DUF283 facilitates the hybridization between the opposite strands of nucleic acids [48]. Moreover, dsRNAs regulate the protein nucleo-cytoplasm distribution and bind with dsRNAs [49,50].

Dicer proteins are conserved in different plant species. Four main classes of DCL have been reported up till now. DCL1 is involved in generating and processing miRNAs and 21/22 nt sRNAs from short hairpins [51,52,53]. DCL2 results in the production of 22 nt sRNAs from long hairpin RNAs, viral dsRNA processing, the modification of chromatin, and production of heterochromatin siRNAs [54,55]. DCL3 plays the same functions as DCL2, as it generates 24 nt sRNAs. DCL4 processes long dsRNA and trans-acting small interfering RNAs (ta-siRNAs) to produce 21 nt sRNA [50,51,52]. Additionally, a crucial function is the generation of siRNAs from dsRNAs during an RNA-directed DNA methylation (RdDM) mechanism [56,57,58,59].

#### 3.1.2. RNA-Induced Silencing Complex (RISC)

The RISC is the complex of ribonuclease proteins that degrade the targeted mRNA through sequence-specific nucleases. The RISC is a 250–500 kDa precursor complex that is activated with ATP into a 100 kDa activation complex [60,61]. The siRNA is a vital member of the RISC, guiding the complex to bind with targeted mRNA. Different protein components of the RISC have been identified, and a significant one is the Argonaute (AGO) protein. AGOs are about 100 kDa proteins that comprise three components: the N-terminal PAZ domain, while the P-element induced wimpy (PIWI) domain is in the mid and C-terminal [62]. The PAZ domain is required for the protein–protein interaction through its RNA-binding module. However, PIWI is required for the cleavage of targeted mRNA [63]. AGO proteins in plants are involved in DNA methylation, gene silencing, plant growth, and defense or immune response [32,64,65,66]. AGO 4, 6, and 9 participate in transcriptional gene silencing (TGS), while AGO 1, 2, 3, 5, 7, and 10 play vital roles in post-transcriptional gene silencing (PTGS) [67]. AGO proteins are also involved in RNA-dependent DNA methylation (RdDM) pathways [68].

#### 3.1.3. RNA-Dependent RNA Polymerase (RDR)

RDR is involved in the continuation of the RNAi process through the amplification of sRNA. This enzyme amplifies the single-stranded RNA molecules and converts them into a dsRNA form, cleaving them by DCLs to generate new sRNAs [69]. A crucial role of RDR proteins is to interact with RNAi machinery and provide defense against pathogens. RDR proteins in plants are identified as RDRγ [70], which is evolutionarily distributed into four subclasses (RDR1, 2, 3, and 6) [71,72]. Among them, RDR1, RDR2, and RDR6 are vital proteins that are involved in the defense against pathogens [73,74,75], the biosynthesis of siRNAs [54,76], DNA methylation, and epigenetic silencing [77].

### 3.2. Different Types of Gene Silencing

#### 3.2.1. Virus-Induced Gene Silencing (VIGS)

In virus-induced gene silencing (VIGS), the viral genome is manipulated by deleting the disease-causing genes and then the cloning of the modified viral genome cDNA into a binary vector. Viruses that lack gene silencing or weak suppressors are the potential target as VIGS vectors [78,79,80,81,82,83]. The targeted vector also contained multiple cloning sites (MCS) which can insert targeted gene fragments [78,79,80]. The specific silenced gene is cloned into the MCS of the binary vector [81]. The recombinant virus then enters the plant cells through an Agrobacterium-mediated transformation or DNA bombardment into the host cells. Once the recombinant virus enters the plant cell, RNA-dependent RNA polymerase (RdRp) transcribes the viral RNA and transgene [82]. Double-stranded RNAs (dsRNAs) are generated and further cleaved by Dicer into 21–25 nts long siRNA. These siRNAs further loaded into the RISC that targeted the complementary DNA [32,83].

More than 35 DNA or RNA viruses have been altered as VIGS vectors [84]. The *Tobacco mosaic virus* (TMV), *Tomato golden mosaic virus* (TGMV), and *Potato virus* X (PVX) belong to the first generation of the VIGS vector system. The first-generation VIGS vector caused the short-term silencing of endogenous gene expression and leaf chlorosis [85,86,87]. For the second-generation VIGS vectors, viruses responsible for milder symptoms were selected. Improvements were made in the targeted VIGS vectors, such as deleting pathogenicity-related genes and adding inverted repeats for hairpin RNA formation [88,89]. These alterations result in fewer side effects in the host plants. Additionally, the first-generation VIGS vectors have a limited range of host susceptibility, except for the *Bamboo mosaic virus* (BaMV), which has a broad host range [90]. However, the most commonly used VIGS vector is the *Tobacco rattle virus* (TRV), which has the ability of a wide host range infection, with its systemic transmission including the meristem and minor virus infections in infected plants [91,92].

#### 3.2.2. Host Induced Gene Silencing (HIGS)

Host-induced gene silencing is based on the plant’s natural immune system, which utilizes RNA-induced silencing to defend the viral infection [93,94,95]. HIGS is further advanced to VIGS, which silences the pathogenic genes inside plants by expressing RNAi constructs, targeting the specific genes of the pathogen inside the host plant [94]. In HIGS, transgenic plants are generated by introducing an inverted repeat sequence inside the plant genome. Double-stranded RNAs are produced as small RNAs inside the transgenic plants, introduced either through Agrobacterium or VIGS. A homologous sequence to the gene or genes from pathogens, which is incorporated into plant genomes, expresses the siRNA of targeted genes and leads to the pathogen’s gene silencing [96].

Previously, applying chemicals to virus-infected plants was the only source to limit virus infections. However, with the advent of RNAi technology, it has been established that HIGS is a powerful and efficient technology to control the virus spread and infection inside the plant cell. The development of efficient, resistant, and polycistronic miRNA and the fusion of multiple genes in hairpin RNA is efficiently successful [97,98]. Noticeably, infections against the *Wheat streak mosaic virus* (WSMV) were reduced using a coat protein and a full-length viral replicase (NIb) gene [99,100,101].

#### 3.2.3. Spray Induced Gene Silencing (SIGS)

Spray-induced gene silencing (SIGS) is an additional advanced RNA silencing strategy for disease control. SIGS has been effectively used for monocots and dicots pathogen infections [96]. SIGS has been used for crop protection based on findings that plant pathogens can up-take dsRNA, which is applied externally [102,103]. This dsRNA then silences the targeted pathogen genes which is critical for disease improvement. SIGS is an eco-friendly and advanced strategy for pathogen control at pre-harvesting and post-harvesting stages and offers fewer off-target effects [104]. The topical application of dsRNA confers resistance against *Alfalfa mosaic virus* (AMV), *Pepper mild mottle virus* (PMMoV), and *Tobacco etches virus* (TEV) [105]. More research has been carried out using dsRNA, which provides resistance against various host plants’ viral infections [106,107,108,109].

Previously, an RNA-based strategy targeting plant transgenes had been reported, mainly in *Nicotiana benthamiana*, Arabidopsis, and *Oryza sativa*. Carborundum/Silwet L-77 surfactant was sprayed on *N. benthamiana* leaves before 21 nt sRNAs targeted the *PHYTOENE DESATURASE* (*PDS*) gene [110]. The exogenous application of dsRNA included syringe infiltration, irrigation methods, foliar spray, and an adjuvant with formulations. 

Syringe infiltration is one of the simplest methods for transgene introduction in experimental plants. It was observed that dsRNA infiltration into leaf targeting the YFP transgene in Arabidopsis failed to develop RNAi. However, the introduction of dsRNA and a carrier complex constituting a co-polymer of lysine or histidine enhanced the efficacy of dsRNA delivery, which targeted the Chalcone Synthase Gene (CHS) and the YFP transgene, resulting in the decreased accumulation of targeted genes [111].

Mechanical inoculation with soft sterile brushes also facilitates the introduction of dsRNA. Additionally, gentle rubbing of the inoculum at the center of the leaf was also reported to deliver dsRNA to induce RNAi. dsRNAs targeting Neomycin phosphotransferase II (NPTII) and eGFP were transferred into transgenic Arabidopsis plants through soft brushes, reducing target genes and their protein levels [107]. Furthermore, the inoculation of dsRNA targeting the MYB1 transcription factor (TF) through gentle rubbing results in flattened epidermal cells instead of conical cells on the Dendrobium hybrid orchid plant [112]. However, the rub inoculation and infiltration method are unsuitable for greenhouse or field conditions. These constraints were overcome through the spray-based inoculation method. SIGS had been performed by the low-pressure spraying of siRNAs, which was previously reported to fail the GFP silencing. However, the problem was overcome by spraying siRNA at a high pressure, which successfully silenced the GFP. Further investigation is required to optimize the high-pressure spraying technology against targeted tissues [104].

#### 3.2.4. Advantages of SIGS over HIGS and VIGS

RNAi is responsible for targeted mRNA removal from the transcriptome. It is suitable to modulate those plant traits that negatively impact the plant phenotype. For example, in the case of a plant-pathogen infection, the pathogen delivers effector proteins into infected plant tissue to hijack host proteins and suppress the host immune system. Effector proteins inside the target host cells critical for the host–pathogen interaction are often known as susceptibility factors [113].

RNAi mediated a broad range of biological processes such as plant growth, development, and the host–pathogen interaction [102,114]. HIGS provides the silencing of the targeted genes of pests and pathogens by generating transgenic plants [115]. However, using transgenes and genetically modified organisms (GMOs) in HIGS and VIGS may raise consumer concerns. However, SIGS is advantageous over other RNAi technologies as it does not require genetic modification. It is also beneficial, especially for those crops that are difficult to modify through transgene and gene editing. Therefore, SIGS is an eco-friendly strategy that can control plant diseases at the pre-harvest and post-harvest stages.

### 3.3. Gene Silencing for Plant Virus Resistance

Plant viruses are a significant threat to crops and cause approximately 10–15% of economic loss [8]. Plant viruses enter host plants through wounds or insect vectors, then multiply inside host cells, causing a transient and systemic spread [116]. Plants use various mechanisms to limit virus infection, including hypersensitive response (HR), systemic acquired resistance, and DNA methylation [100]. Virus resistance has been engineered using small RNA-based strategies such as artificial microRNA (amiRNA) and hairpin RNA (hpRNA) [117,118,119].

RNA gene silencing has been effectively used to target economically significant plant viruses, such as *Mungbean yellow mosaic India virus* (MYMIV) [120], *Cassava mosaic virus* (CMV) [121], *Papaya ring spot virus* (PRSV) [122,123], *Citrus tristeza virus* (CTV) [124], *Maize streak virus* (MSV) [125], *Maize dwarf virus* (MDV) [126], *Soyabean mosaic virus* (SMV) [127], *Potato leaf roll virus* (PLRV) [128], *Zucchini yellow mosaic virus* (ZYMV) [128], *Plum pox virus* (PPV) [129], *Prunus necrotic ringspot virus* (PNRV) [130], *Banana bract mosaic virus* (BBrMV), *Banana bunchy top virus* (BBTV) [131], *Cucumber mosaic virus* (CMV) [132], *Potato virus* X (PVX) [133], *Watermelon silver mottle virus* (WSMV), *Tomato leaf curl burewala virus* (TLCBV), *Tomato leaf curl Delhi virus* (TLCDV), *Wheat streak mosaic virus* (WSMV) [100], *Barley yellow dwarf virus* (BYDV) [134], *Potato virus* Y (PVY) [133], *Tomato yellow leaf curl virus* (TYLCV) [135], *Sugarcane mosaic virus* (SCMV) [136], and *rice tungro bacilliform virus* (RTBV) [137].

### 3.4. RNAi against Viral Suppressors

Viral suppressors are encoded by most plant viruses, which inhibit the RNAi defense system of the infected plant. An RNAi-based strategy is used to control viral infections by inactivating viral suppressors on several proteins. These proteins include V2 in the *Tomato yellow leaf curl virus* (TYLCCNV) and *yellow leaf curl virus* (TYLC), C2 of Curtoviruses, AC2 of Begomoviruses, betasatellites of TYLCCNV, the helper component proteinase of potyviruses of the *Tobacco etch virus* (TEV), *Sugarcane mosaic virus* (SCMV), and *Turnip mosaic virus* (TMV), P38 of *Turnip crinkle virus*, 2b of *Cucumber mosaic virus* (CMV), and P6 of *Olive mild mosaic virus* (OMMV) [138,139,140,141,142,143,144].

### 3.5. RNAi against Insect Vectors

Most plant viruses are transmitted to the host plants through insect vectors such as aphids. Therefore, controlling aphid vectors is another alternative means to block the virus spread. Virus transmission is acquired by an aphids–virus protein interaction through receptors distributed on the salivary glands and gut of the aphid vector [145]. This receptor–protein interaction is a potential target for controlling virus transmission through RNAi [146]. Similarly, cuticular proteins control the virus and vector interaction, enter the gut, and protect the virus degradation in insect hemolymph. A cuticular protein receptor in the acrostylet of the pea aphid is responsible for virus transmission in a circulative manner [147]. Additional research was accomplished by silencing the Rack1 gene of the midgut found in a green peach aphid which reduced nutrient uptake [148]. Previously, C002, an effector protein of the salivary gland, was selected as a suitable candidate for RNAi. This protein plays a crucial role in suppressing the plant defense system and detoxifying plant secondary metabolites [149]. Enzymes of MIF1, glutathione-S-transferase-1, and Armet, are suitable targets for RNAi and are involved in the feeding and growth of aphids [150,151]. Besides salivary proteins, other aphid enzymes are also essential targets for RNAi. For example, the alteration of the fatty acid biochemical pathway increased the aphids’ mortality and decreased its fecundity [152].

### 3.6. HIGS against Plants Viruses

HIGS has been proven to be an efficient technology against plant virus resistance development. Previously, transgenic wheat lines were generated against the *Wheat streak mosaic virus* (WSMV) using a viral replicase (Nib) full-length gene [97,101]. Transgenic lines harboring hairpin RNA transgenes showed efficient resistance against WSMV (T5 generations). Moreover, HIGS was reported against the *Triticum mosaic virus* (TriMV), in which hairpin constructs were generated through the TriMV CP gene, which showed stable transgene resistance through T6 [153]. Additionally, RNAi-mediated transgene resistance was acquired against the *Sugarcane mosaic virus* (SCMV), which was accomplished by constructing hairpin RNA sequences against the Coat protein (CP) and Helper-component proteinase (Hc-Pro) genes. The fusion of both genes developed stable transgenic resistance against SCMV in a model rice plant [101].

### 3.7. VIGS against Plant Viruses

VIGS is a fast and efficient method to analyze gene functions in plants [89]. VIGS vectors can be designed from RNA and DNA viruses to silence targeted genes inside plant cells [154]. By far, TRV has a broad host range, such as *Zea maize* [155], *Nicotiana benthamiana* [156], *Arabidopsis thaliana* [156], *Hyoscyamus niger* [157], and *Gerbera hybrids* [158]. Another virus, the *Apple latent spherical virus* (ALSV), is also utilized as a VIGS vector with a broad spectrum. Up till now, ALSV has been used for several crops, such as *Nicotiana* spp., *A. thaliana*, *S. lycopersicum*, legumes, cucurbits, Malus, and *Pyrus* spp. [159,160]. Furthermore, *the Citrus leaf blotch virus* (CLBV) has been developed as a VIGS vector for *N. benthamiana* and *Citrus* [161]. The *Cucumber mosaic virus* (CMV) is an efficient vector to down-regulate the anthocyanin content in Glycine max by the silencing of the *CHALCONE SYNTHASE* (CHS) gene [162]. The *Barley stripe virus* (BSMV) and *Foxtail mosaic virus* (FoMV) are the vectors for monocots, previously used to silence the PHYTOENE DESATURASE (PDS) gene in *Hordeum vulgare* and *Zea mays*, respectively [163,164].

### 3.8. SIGS against Plant Viruses

In recent years, SIGS has been in practice to reduce the number of diseases caused by pathogens such as fungus, bacteria, and viruses [106,108,109]. SIGS was firstly reported to be against the *Pepper mild mottle virus* (PMMoV), *Tobacco etch virus* (TEV), and *Alfalfa mosaic virus* (AMV). The dsRNA-targeted viral replicase gene downregulated the viral infection in the infected host plant [105]. The effect of SIGS against various viruses is still under investigation [108].

## 4. Limitations of Gene Silencing

The major limitation of RNAi is the lengthy and expensive procedure, as it requires the generation of transgenic plants. These genetically modified plants also required significant regulations and public awareness issues. There are certain limitations in VIGS technology, despite its numerous advantages. VIGS most often results in random gene silencing throughout the infected plant, which could result in interpretations when the silencing is not related to the visible phenotype. The problem might be solved by adding a positive control in the VIGS vectors that facilitates the visibility of the silenced genes [89]. Moreover, VIGS is responsible for silencing non-target genes that are difficult to understand, especially when the genome sequence of the studied species is not available. This unintentional silencing will be solved as whole-genome sequences for most species become available [89]. HIGS technology has improved plant genomic content for resistance to various plant diseases. However, silencing the individual gene from the pathogen may not be enough to limit the disease, as partial silencing of mRNA does not assure the inactivation of the protein. This problem is solved by initially screening the silencing constructs through the transient system [96]. Another limitation of HIGS is the off-target effects that could be avoided using bioinformatic tools to screen off-targets. SIGS is another advanced and innovative technology to silence pathogenic genes, but the detailed mechanism of RNA molecule transportation is still under investigation [165].

## 5. Conclusions and Future Prospects

RNAi is an effective and robust technique to develop transgenic plants which are highly resistant to virus infection. RNAi technology offers avenues to generate a high-yield and resistant transgenic varieties. Recently, SIGS has emerged as an efficient strategy to develop virus resistance, although much work is underway to optimize SIGS technology against different plant viruses. With more research and scientific progress, all the limitations could be resolved in the future.

## Figures and Tables

**Figure 1 ijms-23-05312-f001:**
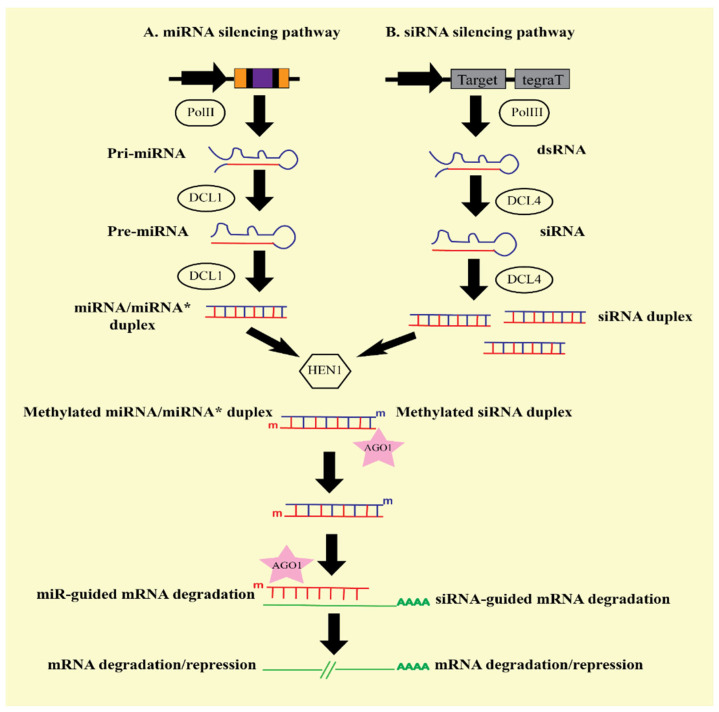
Schematic representation of gene silencing mechanism inside the plant cell. (**A**) Gene silencing pathway via miRNA. miRNA gene is transcribed by RNA polymerase II. It results in the formation of stem-loop structure, which is termed primary microRNA/pri-miRNA. Dicer enzyme (DCL1) cleaved the stem-loop portion in two steps, forming precursor-microRNA (pre-miRNA) and miRNA/miRNA * duplex. The miRNA/miRNA * duplex is methylated by HEN1 protein. Methylated duplex is detected by Agonuate (AGO1) protein of RNA-Induced Silencing Complex (RISC), which unwinds the duplex. Single-stranded AGO1 guided the methylated miRNA strands, which was hybridized with complementary mRNA. Targeted mRNA undergoes cleavage or translational repression. (**B**) Gene silencing pathway via siRNA. The inverted repeat sequence of hairpin RNA is transcribed by RNA polymerase III, which generates a double-stranded RNA (dsRNA) structure. dsRNA is cleaved by DCL4 to generate siRNA duplex. siRNA duplex is methylated by HEN1 protein. AGO1 protein detected the methylated duplex, which underwent unwinding of duplex DNA and complementary paired with targeted mRNA. It also results in the degradation/repression of the target gene [38,39,40,41,42,43].

## Data Availability

The authors confirm that the data supporting the findings of this study are available within the article.

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
