# Peer review of "RNA Interference: Promising Approach to Combat Plant Viruses"

_ijms, 2022, doi:10.3390/ijms23105312_

Round 1
Reviewer 1 Report
This is a nice review discussing how RNA interference technology can be used to combat biotic stresses such as plant viruses. I feel a fair amount of it is nothing new. I like the section on spray technology. It would be great if you could expand on this and other novel sections.
Author Response
Thank you for your kind remarks. More detail about spray technology has been added. Page 5; line no. 208-233.
Reviewer 2 Report
I see the problem with every section with the way they are loosely written and the overall quality of the presentation. The manuscript is poorly written, and the data presentation is weak and difficult to interpret. It lacks proper scientific English Language and Grammer use. Which is difficult to interpret; the Figures lack reasonable explanation.
Other comments are below:
Abstract
Abstract is carelessly written authors should incorporate their notable findings and adequately connect with the sentences they choose to correspond.
Introduction
- The introduction section must have a clear hypothesis and significantly develop the second paragraph of your manuscript. Make it more connecting to the problem statement.
- Overall there is the repetition of the information, which could be avoided.
Discussion
- This section should include more information and references related to the relevant and related works.
Figure
- Figure legends are difficult to interpret.
Conclusions
- If possible, restructure and carefully edit the conclusion section and add clear information regarding the most noteworthy findings.
Author Response
I see the problem with every section with the way they are loosely written and the overall quality of the presentation. The manuscript is poorly written, and the data presentation is weak and difficult to interpret. It lacks proper scientific English Language and Grammer use. Which is difficult to interpret; the Figures lack reasonable explanation.
English language and grammar mistakes are critically checked and corrected.
Other comments are below:
Abstract
Abstract is carelessly written. Authors should incorporate their notable findings and adequately connect with the sentences they choose to correspond.
The Abstract has been modified.
Introduction
- The introduction section must have a clear hypothesis and significantly develop the second paragraph of your manuscript. Make it more connecting to the problem statement. Overall there is the repetition of the information, which could be avoided.
Repetitive information has been removed. Such as microRNA and siRNA headings are removed as they are already explained.
Discussion
- This section should include more information and references related to the relevant and related works.
Literature has been added. Page 5; line no 208-233.
Figure
- Figure legends are difficult to interpret.
The explanation is added to the figure legend.
Conclusions
- If possible, restructure and carefully edit the conclusion section and add clear information regarding the most noteworthy findings.
Conclusion has been modified.
Reviewer 3 Report
In this review, the authors described the mechanism of RNAi system and RNAi pathways such as HIGS, VIGS, and SIGS. The contents are suitable for a book chapter rather than a review paper. In addition, the authors propose some approaches to combat plant viruses by RNAi, but it is not sufficient. The numbering of the title is strange because main section is only INTRODUCTION. Main paragraphs and conclusion should be separated from INTRODUCTION section. Furthermore, the authors did not show any idea how RNAi and genome editing are combined for plant disease control in the conclusion section. Thus, it was difficult for the reviewer to understand the new insights on RNAi technology fully.
Author Response
Thank you for your remarkable suggestions. We have added some more literature related to SIGS technology. The numbering of the title has been corrected, and the conclusion is separated from the Introduction section. Conclusion and future work have been improved.
Round 2
Reviewer 2 Report
Thank you for the corrections, but some sections are still weak and require a careful update for the content. Please include more information in every section.
Author Response
Thankyou for your kind remarks.
Content has been updated. Line 88-109; line 202-213; line 224-230.

Reviewer 3 Report
The reviewer found that the authors improved the manuscript well, but I would like to provide some minor points to be revised.
1) L7 They used dagger mark to show author contribution, but I cannot find any authors with the mark in L3.
2) L94 RNA-induced? L212 Host-induced? L223 Spray-induced?
3) Did the authors cited figure 1 in the main text?
Author Response
- L7 They used dagger mark to show author contribution, but I cannot find any authors with the mark in L3.
Corrected
- L94 RNA-induced? L212 Host-induced? L223 Spray-induced?
General mechanism of RNA induced gene silencing has been described.
- Did the authors cite figure 1 in the main text?
Figure has been cited.
Round 3
Reviewer 2 Report
Manuscript can be accepted for publication after a careful English language check.
Author Response
Thanks for your revision. All was revised as comments.
